# Risk of hand and forearm conditions due to vibrating hand-held tools exposure: a retrospective cohort study from Sweden

Malin Zimmerman ●,[1,2] Peter Nilsson ●,[3] Mattias Rydberg,[4,5] Lars Dahlin ●[4,5]

¹Department of Translational Medicine—Hand Surgery, Lund University, Malmö, Sweden
²Department of Orthopedics, Helsingborgs lasarett, Helsingborg, Sweden
³Clinical Sciences, Lund University, Malmö, Sweden
⁴Department of Translational Medicine, Lund University, Malmö, Sweden
⁵Department of Hand Surgery, Skånes universitetssjukhus Malmö, Malmo, Sweden

**Correspondence to**
Dr Lars Dahlin;
lars.dahlin@med.lu.se

## ABSTRACT

**Objectives** The occurrence of hand and forearm disorders related to vibration exposure, adjusted for relevant background factors, is scarcely reported. We analysed the prevalence of such conditions in a large population cohort, stratified by sex, and associations with exposure to vibrating hand-held tools.

**Design** This is a retrospective cohort study.

**Setting** Individuals in the Malmö Diet and Cancer Study cohort (MDCS; inclusion 1991–1996; followed until 2018) were asked, 'does your work involve working with vibrating hand-held tools?' (response: 'not at all', 'some' and 'much'). Data were cross-linked with national registers to identify treatment for carpal tunnel syndrome (CTS), ulnar nerve entrapment (UNE), Dupuytren's disease, trigger finger or first carpometacarpal joint (CMC-1) osteoarthritis (OA). Cox regression models, unadjusted and adjusted (age, sex, prevalent diabetes, smoking, hypertension and alcohol consumption), were performed to analyse the effects of reported vibration exposure.

**Participants** Individuals in the MDCS who had answered the questionnaire on vibration exposure (14 342 out of the originally 30 446 individuals in MDCS) were included in the study.

**Results** In total, 12 220/14 342 individuals (76%) reported 'no' exposure, 1392/14 342 (9%) 'some' and 730/14 342 (5%) 'much' exposure to vibrating hand-held tools. In men, 'much' exposure was independently associated with CTS (HR 1.71 (95% CI 1.11 to 2.62)) and UNE (HR 2.42 (95% CI 1.15 to 5.07)). 'Some' exposure was independently associated with UNE in men (HR 2.10 (95% CI 1.12 to 3.95)). 'Much' exposure was independently associated with trigger finger in women (HR 2.73 (95% CI 1.49 to 4.99)). We found no effect of vibration exposure on Dupuytren's disease or CMC-1 OA. 'Much' vibration exposure predicted any hand and forearm diagnosis in men (HR 1.44 (95% CI 1.08 to 1.80)), but not in women.

**Conclusions** Vibration exposure by hand-held tools increases the risk of developing CTS and UNE and any common hand and forearm conditions in men, whereas women only risk trigger finger and CMC-1 OA. Adjustment for relevant confounders in vibration exposure is crucial.

## STRENGTHS AND LIMITATIONS OF THIS STUDY

⇒ Different hand and forearm conditions, with long-term follow-up, were analysed in the same population-based cohort of men and women, with calculated HR adjusted for relevant confounders.
⇒ Quantification of vibration exposure is not possible in larger cohorts, but the simple grading (ie, 'no', 'some' and 'much') showed an increased risk for some hand and forearm conditions.
⇒ A 41% response rate, which is a limitation, was obtained among the population-based cohort, with no information on vibration exposure in >10% of the individuals.
⇒ Diagnoses were based on International Classification of Diseases(ICD) codes from the National Board of Health and Welfare, including both surgically and not surgically treated cases, but with no individuals from primary care.

and psychological disability from injuries to various tissue components of the upper extremity, such as muscle, nerve, connective tissue and joints.[1–4] Vibration exposure is reported to induce pathological conditions, including carpal tunnel syndrome (CTS),[5 6] ulnar nerve entrapment (UNE) at the wrist or the elbow,[7] Dupuytren's contracture[8] or osteoarthritis (OA) of the first carpometacarpal joint (CMC-1),[7] all of which may severely affect the patient's hand function and ability to work. The most well-known symptoms are related to the blood vessels, where vibration exposure provokes vibration-induced white fingers.[9 10]

Muscle weakness is a clinically common symptom that is described by individuals exposed to hand-held vibrating tools, and structural changes in the muscles have been described both in experimental studies and in human muscle biopsies.[11] In addition, nerve biopsies from the posterior interosseous nerve just proximal to the wrist from patients exposed to vibrating hand-held tools

## INTRODUCTION

Hand-arm vibration syndrome consists of symptoms and functional, social, emotional

indicate that vibration exposure induces neuropathy with substantial axonal loss.[12] Such a neuropathy may be the underlying reason why exposed patients experience CTS-like symptoms or have a propensity to develop CTS.[3] An increased occurrence of OA in joints of the hand related to vibration exposure has been suggested.[13] A less highlighted condition, although questioned to be work related,[14] and linked to exposure to vibrating hand-held tools is Dupuytren's disease with finger contracture affecting the palmar fascia and bending of fingers.[8] However, a genetic component[15] as well as a relation to diabetes and occupation[16] may be more common causes of Dupuytren's disease. Vibration exposure may promote or speed up the development of Dupuytren's contracture in a genetically predisposed individual.

Most previous studies report on one of the hand and forearm conditions, and few have reported the occurrence of all conditions suggested to be related to vibration exposure.

We aimed to analyse the presence of five common hand and forearm disorders in a large population-based cohort, where a proportion of the individuals reported that they were 'not at all' exposed or exposed to vibrating hand-held tools to 'some' or 'much' extent.

## METHODS
### Study population
The population-based register from southern Sweden, the Malmö Diet and Cancer Study (MDCS), which has previously been described,[17 18] was used in this observational study. In brief, 30 446 individuals aged 46–73 years were invited to participate between 1991 and 1996. Initially, individuals were screened for cardiovascular risk factors, undergoing clinical examination and laboratory assessment under fasting conditions. In addition, they responded to several questionnaires. One of the questionnaires contained the question, 'does your work involve working with vibrating hand-held tools?' where the individuals responded, 'not at all', 'some' and 'much'. This questionnaire was answered by 14 342 individuals included in the present study. Hence, we report self-reported vibration exposure. In addition, other relevant variables were noted, including smoking, alcohol consumption, hypertension and diabetes mellitus. Smoking was self-reported at baseline. Alcohol consumption was self-reported for use during the last week at baseline and reported as grams per day. Diabetes data were obtained through linkage with the National Diabetes Register, which contains data from both hospital care and primary healthcare on patients with diabetes. The present or latest job was defined according to the Swedish socioeconomic classification into three groups; employees (including officials/salaried employees), labourers (including labourers in the production of goods or services as well as farmers) or self-employed. Non-participants of the MDCS are best described in the study by Manjer et al,[18] where mortality

was higher in non-participants, which might reflect a larger disease burden and less willingness to participate.

### Endpoint retrieval
In the present study, we used data from national registers managed by the National Board of Health and Welfare to identify whether the included individuals had been treated for the hand and forearm conditions of interest. The used registers include data from hospitals and specialised care settings (outpatient clinics) but not from primary care units. Inpatient care has been registered since 1964, outpatient surgeries are included from 1997 and visits in specialised outpatient clinics are included from 2001. The included hand and forearm conditions were the two most common nerve compression disorders (CTS (International Classification of Diseases (ICD-10) code G56.0) and UNE at the elbow or the wrist (G56.2)), Dupuytren's disease with contracture (M72.0), trigger finger (M65.3) and OA of the first CMC joint (M18.X). Patients were followed until death, emigration or end of study on 31 December 2018, whichever occurred first.

### Statistical analysis
The baseline characteristics are presented as median (25th to 75th percentiles) or proportion of patients n (%). Since the continuous variables (age and alcohol consumption) were not normally distributed, we used non-parametric testing. The Kruskal-Wallis test was used to detect any statistical significance between groups based on vibration exposure for continuous variables. If p value<0.05, a Mann-Whitney U test was used for pairwise individual comparisons between the three groups. For ordinal variables, the $\chi^2$ test was used for group comparisons. Cox regression analysis was performed to obtain HR for risk of development of hand and forearm pathology due to vibration exposure. Time to event was defined as follow-up period from screening to first disease event or to last follow-up date (31 December 2018; years), emigration or death. In the group with 'much' vibration exposure, there were 82% men, hence the results were stratified by sex. The group without vibration exposure was used as a reference group. We also analysed vibration exposure as a dichotomous variable ('no' vibration exposure vs 'any' vibration exposure). The first models were unadjusted. The second models were adjusted for age at baseline, sex, prevalent diabetes before the study started, smoking, hypertension and alcohol consumption. Confounders were chosen based on previous data and the authors' clinical experience on influencing factors for the specific diagnosis[19–21] (see also Discussion). Finally, Kaplan-Meier curves were used to visualise disease-free survival. All statistical analysis was made using IBM SPSS Statistics for Mac V.28, and a two-sided p value<0.05 was considered significant.

### Patient and public involvement
No patients, politicians or the public were involved in elaborating on the present research question or the design of

the study. The obtained results, after publication in the scientific journal, will be disseminated in different ways.

## RESULTS

### Characteristics

In total, 12 220/14 342 individuals (85%) reported that they did not have had 'any' exposure to vibrating hand-held tools, while 1392/14 342 (10%) reported that they had 'some' and 730/14 342 (5%) had 'much' exposure to vibrating hand-held tools. Age was median 57±6 years in all groups. There were more men in the vibration exposed groups ('not at all' 4845/12 220 (40%) vs 1003/1392 (72%) in the 'some' group and 601/730 (82%) in the 'much' group). Most included individuals (7181/14 342; 50%) were employees, 5440/14 342 (38%) were labourers, and 1684/14 342 (12%) were self-employed. Occupational data were missing in 37 individuals. Basic characteristics are presented in online supplemental table 1. Some differences in basic characteristics were observed among men and women related to the extent of reported vibration exposure (table 1).

### CTS and ulnar nerve compression at the elbow or wrist

Among the men, CTS and UNE were more common in the groups with more vibration exposure; however, not being statistically significant for CTS (table 1). In the Cox regression analysis, 'much' vibration exposure was independently associated with CTS (HR 1.71 (95% CI 1.11 to 2.62)) and UNE (HR 2.42 (95% CI 1.15 to 5.07); table 2). 'Any' vibration exposure was independently associated with a greater risk of CTS (HR 1.41 (95% CI 1.02 to 1.95); table 2). 'Some' vibration exposure was independently associated with UNE (HR 2.10 (95% CI 1.12 to 3.95); table 2). Among the women, CTS seemed more common in the group without vibration exposure; however, it was not statistically significant. The Cox regression models for women showed no significant effects of vibration exposure on the risk of developing CTS and UNE (for survival analysis using Kaplan-Meier curves in men and women regarding CTS and UNE, see figure 1A–D).

### Trigger finger

Trigger finger was more common among women with 'much' vibration exposure than women with 'some' or 'no' vibration exposure (table 1). No differences could be seen between the groups of men. 'Much' vibration exposure was independently associated with the development of trigger finger in women (HR 2.72 (1.49–4.96)), but not in men (table 2; for survival analysis using Kaplan-Meier curves see figure 1E,F).

### Dupuytren's contracture

Dupuytren's contracture was equally common among the groups in both sexes, and no significant effect of vibration exposure on the development of Dupuytren's disease with contracture could be demonstrated in the Cox regression

analysis (tables 1 and 2; for survival analysis using Kaplan-Meier curves, see figure 1G,H).

### OA of the first CMC joint

CMC-1 OA was equally common among the groups in both sexes. In women, 'any' vibration exposure was independently associated with the risk of CMC-1 OA (HR 1.96 (1.23–3.12)) in the Cox regression analysis (tables 1 and 2; for survival analysis using Kaplan-Meier curves, see figure 1I,J).

### Any hand and forearm condition

We found similar proportions of any incident hand and forearm condition among all groups (table 1). In the Cox regression analysis, 'much' vibration exposure predicted any hand and forearm condition in men (HR 1.44 (1.12–1.86)) but not in women (HR 1.45 (0.91–2.32)) (table 2; for survival analysis using Kaplan-Meier curves, see figure 1K,L). 'Any' vibration exposure was associated with a greater risk of any hand and forearm condition in men (HR 1.33 (1.10–1.60); table 2).

## DISCUSSION

The present observational study of a large population-based cohort, where the individuals replied to a question concerning exposure to vibrating hand-held tools and graded as 'not at all', 'some' and 'much', shows that hand and forearm conditions in men, such as CTS and UNE, were significantly higher than in unexposed men, particularly among those with 'much' vibration exposure when adjusted for several relevant confounding factors. In addition, any hand and forearm condition were associated with 'any' and 'much' exposure to vibrating hand-held tools. In contrast, in women, there was an association between 'any' exposure to vibrating hand-held tools and CMC-1 OA and an association between 'much' vibration exposure and trigger finger. The presently described hand and forearm conditions are common in the general population, irrespective of vibration exposure, where prevalence figures for CTS, UNE, Dupuytren's contracture, trigger finger and OA of the CMC-1 joint are reported as 1.4%–15%.[20 22 23] Previously published data have mainly included one of the present diagnoses at a time associated to vibration exposure, while the data from the current cohort also allowed the inclusion of all the relevant, and previously discussed, conditions in the hand and forearm associated to exposure to vibrating hand-held tools.[8 24–26]

Risk assessment in vibration exposure of hand-held tools to different hand and forearm disorders may be complex due to the high prevalence of some of the hand and forearm disorders as well as the association with relevant factors and general comorbidities that must be adjusted for in any statistical analysis.[8 13 21 26 27] In this study, we stratified our data on sex due to the relatively high number of male individuals in the cohort, as well attempting to level out any difference in occurrence of

**Table 1** Baseline characteristics of individuals that reported exposure to vibrating hand-held tools as 'not at all', 'some' and 'much' stratified by sex

| | 'Does your work involve working with vibrating hand-held tools?' | | | | | | | |
| --- | --- | --- | --- | --- | --- | --- | --- | --- |
| | Men (n=6448) | | | | Women (n=7892) | | | |
| | 'Not at all', n=4845 | 'Some', n=1003 | 'Much', n=601 | P value | 'Not at all', n=7374 | 'Some', n=389 | 'Much', n=129 | P value |
| Age (years) | 57±6 | 57±6 | 57±6 | 0.67 | 57±6 | 58±6 | 57±6 | 0.079 |
| Current smoking | 1355 (28%) | 355 (35%) | 211 (35%) | **<0.001**\*† | 2103 (29%) | 122 (31%) | 46 (36%) | 0.10 |
| Antihypertensive treatment | 2499 (52%) | 529 (53%) | 329 (55%) | 0.306 | 3264 (44%) | 179 (46%) | 44 (34%) | 0.053 |
| Prevalent diabetes | 269 (6%) | 69 (7%) | 33 (6%) | 0.249 | 277 (4%) | 16 (4%) | 4 (3%) | 0.87 |
| Alcohol consumption (g/day) | 12.2 (4.5–23.0) | 9.6 (2.9–21.3) | 10.6 (1.8–20.1) | **<0.001**\*† | 5.4 (0.8–11.2) | 4.4 (0.46–10.0) | 4.1 (0.0–10.3) | **0.019**\* |
| Labourers (n, %) | 1234 (25%) | 630 (63%) | 472 (79%) | **<0.001**\*† | 2764 (37%) | 263 (68%) | 77 (60%) | **<0.001**\*† |
| CTS | 133 (2.7%) | 34 (3.4%) | 26 (4.3%) | 0.072 | 327 (4.4%) | 9 (2.3%) | 3 (2.3%) | 0.071 |
| Incident UNE at elbow/wrist | 33 (0.7%) | 14 (1.4%) | 9 (1.5%) | **0.019**† | 51 (0.7%) | 2 (0.5%) | 2 (1.6%) | 0.46 |
| Incident Dupuytren's contracture | 158 (3.3) | 25 (2.5%) | 16 (2.7%) | 0.36 | 87 (1.2) | 7 (1.8%) | 2 (1.6%) | 0.52 |
| Incident trigger finger | 137 (2.8%) | 28 (2.8%) | 23 (3.8%) | 0.38 | 262 (3.6%) | 13 (3.3%) | 11 (8.5%) | **0.011**†‡ |
| Incident osteoarthritis of the thumb CMC-1 joint | 61 (1.3%) | 19 (1.9%) | 11 (1.8%) | 0.20 | 236 (3.2%) | 14 (3.6%) | 6 (4.7%) | 0.60 |
| Any incident hand and forearm disorder | 442 (9.1%) | 94 (9.4%) | 69 (11.5%) | 0.17 | 793 (10.8%) | 41 (10.5%) | 18 (14%) | 0.50 |

P value for good comparisons between the subjects are based on Kruskal-Wallis and Mann-Whitney U test as a post hoc test and $\chi^2$ test for ordinal variables.
Significant values marked in bold.
\*Statistical significance was found between 'not at all' and 'some'.
†Statistical significance was found between 'not at all' and 'much'.
‡Statistical significance found between 'some' and 'much'.
CMC-1, first carpometacarpal joint; CTS, carpal tunnel syndrome; UNE, ulnar nerve entrapment.

**Table 2** Cox regression models—unadjusted and adjusted for relevant factors—with the association between vibration exposure and risk for development of five specific hand and forearm conditions and any of the hand and forearm conditions stratified by sex

| | CTS | UNE | Trigger finger | Dupuytren's contracture | CMC-1 osteoarthritis | Any hand and forearm condition |
|---|---|---|---|---|---|---|
| **Men** | | | | | | |
| **Model 1—unadjusted** | | | | | | |
| 'No' vibration exposure | Reference | | | | | |
| 'Any' vibration exposure | 1.43 (1.04–1.97) | 1.18 (0.69–2.04) | 1.22 (0.88–1.70) | 1.10 (0.78–1.57) | 1.27 (0.82–1.97) | 1.30 (1.08–1.55) |
| 'Some' vibration exposure | 1.21 (0.82–1.79) | 2.20 (1.17–4.12) | 1.01 (0.67–1.53) | 0.78 (0.51–1.20) | 1.57 (0.94–2.64) | 1.05 (0.84–1.32) |
| 'Much' vibration exposure | 1.71 (1.11–2.62) | 2.44 (1.17–5.12) | 1.50 (0.97–2.34) | 0.90 (0.54–1.50) | 1.55 (0.82–2.95) | 1.40 (1.08–1.80) |
| **Model 2—adjusted for age, smoking, alcohol consumption, hypertension and diabetes** | | | | | | |
| 'No' vibration exposure | Reference | | | | | |
| 'Any' vibration exposure | 1.41 (1.02–1.95) | 0.84 (0.44–1.60) | 1.22 (0.88–1.70) | 1.15 (0.81–1.65) | 1.56 (0.97–2.50) | 1.33 (1.10–1.60) |
| 'Some' vibration exposure | 1.18 (0.80–1.76) | 2.10 (1.12–3.95) | 1.01 (0.67–1.53) | 0.82 (0.53–1.27) | 1.55 (0.92–2.61) | 1.06 (0.84–1.33) |
| 'Much' vibration exposure | 1.71 (1.11–2.62) | 2.42 (1.15–5.07) | 1.54 (0.99–2.39) | 0.96 (0.57–1.61) | 1.55 (0.81–2.94) | 1.44 (1.12–1.86) |
| **Women** | | | | | | |
| **Model 1—unadjusted** | | | | | | |
| 'No' vibration exposure | Reference | | | | | |
| 'Any' vibration exposure | 0.85 (0.48–1.52) | 0.89 (0.32–2.50) | 1.08 (0.71–1.64) | 0.85 (0.43–1.70) | 2.06 (1.29–3.27) | 1.12 (0.86–1.46) |
| 'Some' vibration exposure | 0.56 (0.29–1.08) | 0.83 (0.20–3.40) | 1.02 (0.59–1.78) | 1.67 (0.77–3.60) | 1.20 (0.70–2.06) | 1.06 (0.77–1.45) |
| 'Much' vibration exposure | 0.56 (0.18–1.75) | 2.50 (0.61–10.26) | 2.72 (1.49–4.96) | 1.43 (0.35–5.83) | 1.55 (0.69–3.48) | 1.43 (0.89–2.28) |
| **Model 2—adjusted for age, smoking, alcohol consumption, hypertension and diabetes** | | | | | | |
| 'No' vibration exposure | Reference | | | | | |
| 'Any' vibration exposure | 0.86 (0.48–1.54) | 0.78 (0.25–2.44) | 1.06 (0.70–1.63) | 0.71 (0.34–1.51) | 1.96 (1.23–3.12) | 1.09 (0.83–1.42) |
| 'Some' vibration exposure | 0.56 (0.29–1.09) | 0.82 (0.20–3.36) | 1.05 (0.60–1.83) | 1.76 (0.81–3.80) | 1.19 (0.69–2.03) | 1.07 (0.78–1.46) |
| 'Much' vibration exposure | 0.56 (0.18–1.76) | 2.54 (0.62–10.48) | 2.73 (1.49–4.99) | 1.57 (0.39–6.38) | 1.57 (0.70–3.54) | 1.45 (0.91–2.32) |

CMC-1, first carpometacarpal joint; CTS, carpal tunnel syndrome; UNE, ulnar nerve entrapment at elbow/wrist.

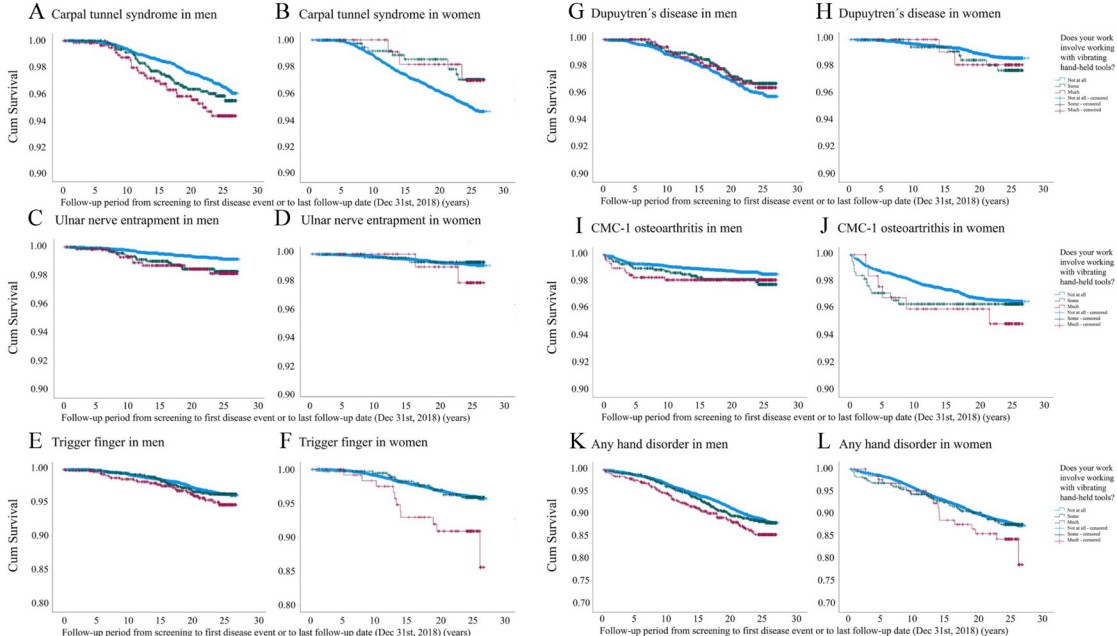

**Figure 1** Survival analysis using Kaplan-Meier curves of the studied hand and forearm conditions that were followed from screening to the first disease event or to the last follow-up date (31 December 2018) and were potentially associated with vibration exposure stratified by sex. The individuals replied to the question 'does your work involve working with vibrating hand-held tools?' with possible answers 'not at all', 'some' and 'much'. CMC-1, first carpometacarpal joint.

vibration exposure in men and women as sex is known to influence the prevalence of hand and forearm diagnoses.[28] In our model 2, we adjusted for factors that may be of relevance in development of the included hand and forearm disorders, that is, age, smoking, alcohol consumption, hypertension and diabetes; variables that are accessible in the MDCS database.[19 27] Age is a highly relevant influencing factor for development of hand and forearm disorders, where men and women have different age distributions for development of CTS and UNE as well as of OA in the first CMC joint.[23 28 29] In contrast to our study, other authors have reported an increased risk for CTS in both sexes, although being highest among young men,[24] which may be explained by differences between the cohorts and adjustment for confounding factors. Smoking may affect both the development and outcome of surgery of CTS and UNE,[27] due to its effect on the intraneural microcirculation as well as being highlighted in Dupuytren's disease with contracture.[30] Individuals with hypertension may also have a higher perfusion pressure in the intraneural blood vessels, making the nerve more resistant to compression[31]; again, indicating that adjustment for several factors in the Cox analysis is necessary.[32] Development of CTS, UNE, Dupuytren's disease with finger contracture and trigger fingers are also associated with both types 1 and 2 diabetes,[19] which is probably not entirely true for OA of the first CMC joint, for which high BMI and obesity are significant risk markers/factors.[33] Finally, alcohol consumption is also a factor that may be involved in development of Dupuytren's contracture as well as a cause of UNE.[19 34] Thus, it is crucial to adjust for age, smoking, alcohol consumption, hypertension and

diabetes when analysing the risk of developing various hand and forearm disorders in relation to vibration exposure of hand-held tools. We also included the association of any hand and forearm disorder as a variable in relation to vibration exposure, which is appropriate since there usually is an extensive comorbidity of various hand and forearm disorders, such as concomitant CTS in surgically treated patients with UNE, as well as the other present hand and forearm conditions.[35] In addition, a subject with diabetes, irrespective of having type 1 or type 2 diabetes or if neuropathy is present, may have an increased susceptibility to nerve compression disorders as a subject with a history of vibration exposure.

The observed differences in risks for developing the studied hand and forearm conditions in men and women are interesting and may have different explanations. Surprisingly, in women, 'much' vibration exposure only increased the risk of trigger finger, while in men, the risks were higher for CTS and UNE. Women also differed in the marked risk for CMC-1 OA with 'any' vibration exposure. There are no apparent reasons for this discrepancy. Still, it has been suggested that an underlying neuropathy may be important for development of CTS in vibration-exposed individuals.[3 8 12] One other study, however, found that working with vibrating tools for more than 2 hours per day was a risk factor for developing CTS in women but not in men.[36] Furthermore, neuropathy is reported to be more common and more severe in men with diabetes.[24 37] Therefore, one may speculate that similar mechanisms may explain the present sex difference concerning nerve compression disorders. Trigger finger is more common among women, and it has been speculated that oestrogen

deficiency alters tendon metabolism in postmenopausal women, hence impacting the risk of developing trigger finger.[20] Previous studies have failed to demonstrate an association between occupational factors and trigger finger.[20] It is, however, possible that during the early 1990s, when this cohort study started, occupational sex differences of risk existed, and among the participants who worked with vibrating hand-held tools, men worked with large tools, such as grinding machines and chainsaws. In contrast, women may have worked more with precision dental drills and other smaller tools. Whether the type of tool affects these disorders' development is unknown.

Our data also indicate that there is a greater risk of OA in the first CMC joint in vibration-exposed women. In contrast, one older study investigated radiological OA in vibration exposed and non-exposed individuals, and found no significant differences between groups, indicating that the development of such a condition has other causes than vibration exposure alone.[25] It is, however, unclear whether that study included any women. One more recent meta-analysis could not demonstrate any increased risk of hand OA from vibration exposure.[13] However, women have been under-represented in many studies on vibration exposure. There may also be a possibility that women may be exposed to other magnitudes and frequencies of vibrations like from various precision instruments requiring a firm pinch grip, such as dental instruments, as indicated by the present higher proportion of labourers among men with 'much' vibration exposure compared with the lower proportion of labourers among women with 'much' vibration exposure.[38–41] Tasks requiring pinch strength of higher magnitude have been associated with an increased risk of hand OA.[13]

It was not possible and was not the present study's purpose to evaluate any surgery outcome for the conditions in the individuals with 'no', 'some' or 'much' exposure to vibration. The result of surgery in vibration-exposed patients with CTS seems to be impaired compared with otherwise healthy patients with CTS.[42] Still, outcome data are essentially lacking for the other hand and forearm conditions included in the present study, except for Dupuytren's contracture, for which socioeconomic status has to be considered.[43] Earlier exposure to vibrating tools may affect the pre- and postoperative function in individuals undergoing surgery for Dupuytren's contracture. However, the satisfaction or return to work was not different.[44] Hence, whether vibration exposure affects clinical outcomes following treatment for common hand and forearm conditions remains an important area for future research.

### Limitations

There are some limitations of the study, such as that in >10% of the individuals, no information was available concerning vibration exposure as well as a 41% response rate among the population-based cohort.[18] This might potentially limit the generalisability of this study, and there might exist a selection bias in who answered the questionnaires. However, based on previous information there may be an age difference (ie, younger being non-responders) between those who do respond and do not respond to questionnaires,[45 46] but recent data indicate that the response rate may not be relevant.[47] In addition, our diagnoses were based on ICD codes from the National Board of Health and Welfare, where both surgically and not surgically treated cases were included, although without any patients diagnosed in primary care. This might include a selection bias, with a possible underestimation of the prevalence, since patients with milder forms of the studied hand and forearm conditions that can be treated conservatively are managed in primary care. Quantifying the vibration exposure is relevant but is not possible in larger cohorts as in the present study, which is a limitation. We also had no data on hand dominance, nor ethnicity and socioeconomic deprivation that might influence prevalence of the studied conditions. Access to socioeconomic data may have subanalyses related to different socioeconomic status to be performed. Thus, the present prevalence may be underestimated if only those with manual work of certain severity, low income and low education levels are evaluated. However, analysis of socioeconomic status in relation to vibration exposure was not in the scope of the study, but is a future project.

### Conclusions

We conclude that vibration exposure to hand-held tools increases the risk of having or developing CTS, and particularly UNE, but also any common hand and forearm conditions in men and CMC-1 OA and trigger finger in women. It is essential to adjust for relevant influencing factors, such as age, smoking, alcohol consumption, hypertension and diabetes, when investigating the risks of working with hand-held vibrating tools.

**Acknowledgements** The authors are very grateful to all the participants in MDCS whose participation enabled this work. The authors also thankful to the National Diabetes Register and Statistics Sweden for register data, Anders Dahlin for his help with data extraction and Tina Folker for her administrative assistance.

**Contributors** MZ, PN and LD conceived the ideas. MZ and PN collected the data. MZ, MR and LD analysed the data. MZ and LD led the writing, and all authors contributed to the interpretation and the writing and approved the final version of the manuscript. LD is the guarantor of the overall content and accepts full responsibility of the work, had access to the data and controlled the decision to publish.

**Funding** This research was funded by the Swedish Research Council (#2022-01942, principal investigator LD), the Swedish Diabetes Foundation (#DIA2020-492, LD), the Regional Agreement on Medical Training and Clinical Research (ALF) between Region Skåne and Lund University (#2018-Projekt 0104; LD) and Funds from Skåne University Hospital (#2019-659; LD), Elly Olsson's Foundation for scientific research (not applicable), Stig and Ragna Gorthon Foundation (not applicable), Almroth Foundation (not applicable), Kockska Foundation (not applicable) and the Magnus Bergvall Foundation (#2020-03612, principal investigator MZ). The initial financial support to the Malmö Diet and Cancer Study cohort at baseline was provided by the Swedish Cancer Foundation (not applicable), The Swedish Medical Research Council (not applicable), The European Commission (not applicable), the City of Malmö (not applicable), the Swedish Dairy Association (not applicable) and the Albert Påhlsson Foundation (not applicable). The authors also acknowledge support from the Lund University Infrastructure grant 'Malmö population-based cohorts' (STYR 2019/2046). The sponsors had no role in the

study design, collection, analysis or interpretation of the data, writing the report or decision to submit the manuscript.

**Competing interests**  None declared.

**Patient and public involvement**  Patients and/or the public were not involved in the design, or conduct, or reporting, or dissemination plans of this research.

**Patient consent for publication**  Consent obtained directly from patient(s).

**Ethics approval**  This study involves human participants and was approved by the ethics committee in Lund and the Swedish Ethical Review Authority (DNR: LU51-90; 2009-633; 2019-01439). Participants gave informed consent to participate in the study before taking part.

**Provenance and peer review**  Not commissioned; externally peer reviewed.

**Data availability statement**  The data are not publicly available due to ethical restrictions. The data that support the findings of this study are only available on request from the corresponding author as well as after application and approval by the Swedish Ethical Review Authority (https://etikprovningsmyndigheten.se/en/) due to the restrictions in Sweden, based on the national Law for ethical review of research on humans ('Lag (2003:460) om etikprövning av forskning som avser människor').

**ORCID iDs**
Malin Zimmerman http://orcid.org/0000-0002-9925-3838
Peter Nilsson http://orcid.org/0000-0002-5652-8459
Lars Dahlin http://orcid.org/0000-0003-1334-3099

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
