## [Reviewer comments · BMJ Open]

ARTICLE DETAILS

TITLE (PROVISIONAL)	Risk of hand and forearm conditions due to vibrating hand-held tools exposure – a retrospective cohort study from Sweden
AUTHORS	Zimmerman, Malin; Nilsson, Peter; Rydberg, Mattias; Dahlin, Lars

VERSION 1 – REVIEW

REVIEWER	Barakat, Ahmed Brighton and Sussex University Hospitals NHS Trust, Trauma & Orthopaedics
REVIEW RETURNED	26-Oct-2023

GENERAL COMMENTS	This is a good study on the association between the use of vibrating tools and the occurrence of different hand conditions such as carpal tunnel, ulna nerve entrapment, trigger finger, 1st CMCJ OA, and Dupuytren's contracture. The methods and discussion are clearly presented and read well. However, a limitation of the study which should be added is that exposure and hand dominance were not correlated with the occurrence of the condition in that dominant hand or not. This would have added to the strength of the correlation if investigated. I understand that dealing with such big cohorts and exploratory questionnaires might be challenging to gather this information however this is a limitation that needs to be acknowledged.
---

REVIEWER	Bhashyam, Abhiram Harvard Combined Orthopaedics Residency Program, Orthopaedic Surgery
REVIEW RETURNED	31-Oct-2023

GENERAL COMMENTS	Thank you for the opportunity to review this paper. This was well executed and interesting. 1) is it possible to assess vibration exposure as a binary variable? I wonder if this may ameliorate response uncertainty between some/much groups. 2) are worker occupations present? Is it possible to account for the duration of exposure?
--

REVIEWER	Buhler, Miranda University of Otago, School of Physiotherapy
REVIEW RETURNED	21-Nov-2023

GENERAL COMMENTS	This manuscript reports a retrospective study of a cohort in an earlier epidemiological study that re-analyses selected questions to examine the association between vibration exposure by hand-held tools and development of carpal tunnel syndrome, trigger
---

digit, ulnar nerve neuropathy at the elbow, Dupuytren's disease, and thumb carpometacarpal osteoarthritis.

This is a well conducted study that uses appropriate statistical methods to produce hazard ratios. It finds that vibration exposure increases the risk of carpal tunnel syndrome and ulna nerve neuropathy in men, and trigger finger in women. The manuscript is well written. This is an important study that should have a significant impact on the design and delivery of health prevention interventions, and on decisions by funders/insurers to attribute causation to specific work roles.

Most comments given below are minor. The most significant point I would offer is: Ethnicity and possibly Socioeconomic deprivation would be two key factors that may have a bearing on the onset of the included conditions and may confound the results. If these data are available it would be valuable to include in the analysis. If not, some well-structured discussion should be added in the Limitations.

Additional comments:

- Suggestion for the title: Risk of hand and forearm conditions due to vibrating hand-held tool exposure – a retrospective cohort study.
- 'Complications' is not an appropriate way to describe the conditions reported on. Suggest instead 'conditions'. The inclusion of ulna nerve neuropathy at the elbow means 'hand and forearm' is more appropriate than 'hand' alone.
- "no", "much", and "some" are more clearly understood when given in their apostrophes, as they are such common words. Suggest these are consistently written with apostrophes throughout the entire manuscript. This will require close checking as it is easy to miss where these words are intended as the specific descriptors.
- In the Abstract, hazard ratio will need written in full the first time used.
- The Introduction section should be broken into 2 or more paragraphs.
- In the main Methods section, it would be helpful to mention hazard ratios and where they fit as the main outcome, before being first presented with them in the Results section.
- 'Strength and limitations of this study' section is wordy in places. Suggest delete the wording here in brackets: *(The main strength is that) different hand complications... *but the (used) simple grading... *A (general) 41% response rate....
- In the Methods, 'Diabetes data was obtained to register linkage...' is not clear what this means.
- In Methods, 'The second models were adjusted for... and on clinical experience on relevant influencing factors for the specific diagnosis' – it is not clear what 'clinical experience' is here and how that was introduced into the model. Please explain.
- The Patient and Public Involvement Statement should be much more concise given there is not much to say here. Limit to two lines.
- In the Results, the first paragraph repeats the information given in Table 1, and Table 2 is somewhat repetitive of the information in Table 1 albeit split between the sexes. Could the Table 2 have the addition of a 'total' column so that Table 1 can be done away with?
- The Discussion of why sex differences exist is a good beginning but could be refined to articulate points more clearly.
- In the Discussion, 'The observed differences... in men and women are exciting...' is an inappropriate use of 'exciting'. Perhaps 'interesting' or 'notable'.

	 • In the Discussion, '(It is possible that) among...' – suggest delete in brackets. • In the Discussion, the paragraph discussing surgery outcome requires some refinement and clearer conclusion – it is not quite clear the point being made. • 'General' is not needed for 'response rate' – suggest delete throughout manuscript. • The Discussion /Limitations mentions that secondary but not primary care data on recorded health conditions could be included. Further discussion is needed to detail what, therefore, is missing, and what is the potential direction and magnitude of bias for the results. I.e., how and to what extent does this threaten the data. E.g., as a start, I wonder if it is less likely to see CMC OA in specialise care as surgical management is less desirable and less definitive compared with carpal tunnel, Dupuytren's, and lesser extent ulna nerve neuropathy and trigger digit. • The Limitations mentions a proportion of respondents for whom no information was available about vibration exposure, and a 59% non response rate among the cohort overall. What is the potential implication of these, for direction and magnitude of any bias in results?
--	---

REVIEWER	Boutsikari, Eleni National and Kapodistrian University of Athens, Medicine
REVIEW RETURNED	07-Dec-2023

GENERAL COMMENTS	I would like to commend the authors for the excellent piece of research that provided insights on a field that requires further attention by the scientific community. Please find my comments inline:  1. I suggest reporting the fact that the scale used to measure exposure to vibration is self-reported, as self-reported data are inherent to certain types of bias (i.e. recall bias) and the latent distance between "not at all"- "some" and "some"- "much" does not -presumably- originate from a validated measure, which would make up for much more unbiased estimates. If the above-mentioned scale is anchored to a quantifiable measure (i.e. certain vibration "dosage") or even somehow validated, this should be mentioned as well. 2. In line 26 the sentence that includes the Kruskal-Wallis test should be clarified - "it was used to detect any statistical differences" but amongst which variables and for what purpose? 3. It appears that Kruskal-Wallis preceded Mann-Whitney. Why would a test that assesses differences between groups (Kruskal-Wallis) be performed prior to another test that assesses differences between groups (Mann-Whitney)? It is suggested to use a normality test first (i.e. Kolmogorov-Smirnov) and then select the appropriate test to assess differences, although with big samples parametric tests can be directly used (please see Central Limit Theorem). 3. Lines 34-36: It is mentioned that Cox modelling was used to analyze the effect of vibration on the development of hand pathology - this implies a yes/no outcome which is typically assessed by logistic regression. Kindly correct the wording.
---

VERSION 1 – AUTHOR RESPONSE

Reviewer: 1

Dr. Ahmed Barakat, Brighton and Sussex University Hospitals NHS Trust

Comments to the Author:

This is a good study on the association between the use of vibrating tools and the occurrence of different hand conditions such as carpal tunnel, ulna nerve entrapment, trigger finger, 1st CMCJ OA, and Dupuytren's contracture. The methods and discussion are clearly presented and read well. However, a limitation of the study which should be added is that exposure and hand dominance were not correlated with the occurrence of the condition in that dominant hand or not. This would have added to the strength of the correlation if investigated. I understand that dealing with such big cohorts and exploratory questionnaires might be challenging to gather this information however this is a limitation that needs to be acknowledged.

Reply: Thank you for your comments. We agree that hand dominance would have been a valuable addition to the data. Unfortunately, this was not included in the questionnaires, and we do not have this information. Thus, we have now added a paragraph on this in the limitations setting.

Reviewer: 2

Dr. Abhiram Bhashyam, Harvard Combined Orthopaedics Residency Program

Comments to the Author:

Thank you for the opportunity to review this paper. This was well executed and interesting.

1) is it possible to assess vibration exposure as a binary variable? I wonder if this may ameliorate response uncertainty between some/much groups.

Reply: Thank you for the suggestion. We have analyzed vibration exposure as a binary variable in the Cox regressions also, please see data added into Table 3. This clarified some of the results. We have now updated the Results section and Discussion accordingly.

2) are worker occupations present? Is it possible to account for the duration of exposure?

Reply: Yes, we have data on worker occupations. Data have been added to Results and also a paragraph in the Discussion. Unfortunately, we have no data to account for the duration of exposure.

Reviewer: 3

Dr. Miranda Buhler, University of Otago

Comments to the Author:

This manuscript reports a retrospective study of a cohort in an earlier epidemiological study that re-analyses selected questions to examine the association between vibration exposure by hand-held tools and development of carpal tunnel syndrome, trigger digit, ulnar nerve neuropathy at the elbow, Dupuytren's disease, and thumb carpometacarpal osteoarthritis.

This is a well conducted study that uses appropriate statistical methods to produce hazard ratios. It finds that vibration exposure increases the risk of carpal tunnel syndrome and ulna nerve neuropathy in men, and trigger finger in women. The manuscript is well written. This is an important study that should have a significant impact on the design and delivery of health prevention interventions, and on decisions by funders/insurers to attribute causation to specific work roles.

Reply: Thank you!

Most comments given below are minor. The most significant point I would offer is: Ethnicity and possibly Socioeconomic deprivation would be two key factors that may have a bearing on the onset of

the included conditions and may confound the results. If these data are available it would be valuable to include in the analysis. If not, some well-structured discussion should be added in the Limitations.

Reply: We agree. The only socioeconomic data that we have access to is on occupation. We have added this in the Results as well as in the Discussion, and a comment in the limitations section.

Additional comments:

- Suggestion for the title: Risk of hand and forearm conditions due to vibrating hand-held tool exposure – a retrospective cohort study.

Reply: We have updated the title.

- ‘Complications’ is not an appropriate way to describe the conditions reported on. Suggest instead ‘conditions’. The inclusion of ulna nerve neuropathy at the elbow means ‘hand and forearm’ is more appropriate than ‘hand’ alone.

Reply: We agree and have updated the manuscript accordingly.

- “no”, “much”, and “some” are more clearly understood when given in their apostrophes, as they are such common words. Suggest these are consistently written with apostrophes throughout the entire manuscript. This will require close checking as it is easy to miss where these words are intended as the specific descriptors.

Reply: We have edited the manuscript accordingly.

- In the Abstract, hazard ratio will need written in full the first time used.

Reply: We have written hazard ratio in full the first time it is used.

- The Introduction section should be broken into 2 or more paragraphs.

Reply: We have broken up the Introduction section into 3 paragraphs.

- In the main Methods section, it would be helpful to mention hazard ratios and where they fit as the main outcome, before being first presented with them in the Results section.

Reply: We have added this in Methods under the “Statistical analysis” section.

- ‘Strength and limitations of this study’ section is wordy in places. Suggest delete the wording here in brackets: *(The main strength is that) different hand complications... *but the (used) simple grading... *A (general) 41% response rate....

Reply: The section has been shortened.

- In the Methods, ‘Diabetes data was obtained to register linkage...’ is not clear what this means.

Reply: We have clarified this in the Methods section.

- In Methods, ‘The second models were adjusted for... and on clinical experience on relevant influencing factors for the specific diagnosis’ – it is not clear what ‘clinical experience’ is here and how that was introduced into the model. Please explain.

Reply: We have clarified this in the Methods.

- The Patient and Public Involvement Statement should be much more concise given there is not much to say here. Limit to two lines.

Reply: We have shortened the statement.

- In the Results, the first paragraph repeats the information given in Table 1, and Table 2 is somewhat repetitive of the information in Table 1 albeit split between the sexes. Could the Table 2 have the addition of a 'total' column so that Table 1 can be done away with?

Reply: Yes, we have added the valuable data from Table 1 in the text and instead focused on Table 2.

- The Discussion of why sex differences exist is a good beginning but could be refined to articulate points more clearly.

Reply: We have elaborated on the discussion on sex differences.

- In the Discussion, 'The observed differences... in men and women are exciting...' is an inappropriate use of 'exciting'. Perhaps 'interesting' or 'notable'.

Reply: We have changed "exciting" to "interesting".

- In the Discussion, '(It is possible that) among...' – suggest delete in brackets.

Reply: We have changed the phrasing.

- In the Discussion, the paragraph discussing surgery outcome requires some refinement and clearer conclusion – it is not quite clear the point being made.

Reply: We have now clarified this.

- 'General' is not needed for 'response rate' – suggest delete throughout manuscript.

Reply: We have deleted this.

- The Discussion /Limitations mentions that secondary but not primary care data on recorded health conditions could be included. Further discussion is needed to detail what, therefore, is missing, and what is the potential direction and magnitude of bias for the results. I.e., how and to what extent does this threaten the data. E.g., as a start, I wonder if it is less likely to see CMC OA in specialise care as surgical management is less desirable and less definitive compared with carpal tunnel, Dupuytren's, and lesser extent ulna nerve neuropathy and trigger digit.

Reply: Correct, we have clarified this in the limitations section.

- The Limitations mentions a proportion of respondents for whom no information was available about vibration exposure, and a 59% non response rate among the cohort overall. What is the potential implication of these, for direction and magnitude of any bias in results?

Reply: The best analysis of non-participants in the MDCS is described in a study by Manjer et al from 2001, we have added this reference to our paper, please see the methods sections. Mortality was higher in non-participants, which might reflect disease burden and willingness to participate.

Reviewer: 4

Ms. Eleni Boutsikari, National and Kapodistrian University of Athens

Comments to the Author:

I would like to commend the authors for the excellent piece of research that provided insights on a field that requires further attention by the scientific community. Please find my comments inline:

1. I suggest reporting the fact that the scale used to measure exposure to vibration is self-reported, as self-reported data are inherent to certain types of bias (i.e. recall bias) and the latent distance between "not at all"- "some" and "some"- "much" does not -presumably- originate from a validated measure, which would make up for much more unbiased estimates. If the above-mentioned scale is anchored to a quantifiable measure (i.e. certain vibration "dosage") or even somehow validated, this should be mentioned as well.

Reply: We have clarified this in the Methods section. Unfortunately, we do not have any quantifiable measure on vibration exposure.

2. In line 26 the sentence that includes the Kruskal-Wallis test should be clarified - "it was used to detect any statistical differences" but amongst which variables and for what purpose?

Reply: We have clarified this in the Statistics section.

3. It appears that Kruskal-Wallis preceded Mann-Whitney. Why would a test that assesses differences between groups (Kruskal-Wallis) be performed prior to another test that assesses differences between groups (Mann-Whitney)? It is suggested to use a normality test first (i.e. Kolmogorov-Smirnov) and then select the appropriate test to assess differences, although with big samples parametric tests can be directly used (please see Central Limit Theorem).

Reply: Our continuous variables were not sufficiently normally distributed. Hence, we chose to use non-parametric testing even though the sample is large. We used Kruskal-Wallis to assess differences between all three groups, but then thought it was of interest to describe between which groups the statistical significance was found, and hence did pairwise post-hoc analysis using Mann-Whitney. We have clarified this in the Statistics section.

3. Lines 34-36: It is mentioned that Cox modelling was used to analyze the effect of vibration on the development of hand pathology - this implies a yes/no outcome which is typically assessed by logistic regression. Kindly correct the wording.

Reply: We used Cox regression to be able to account for time-to-event. This was defined as follow-up period from screening to first disease event or to last follow-up date (2018-12-31) (years), emigration, or death. We have clarified this in the Statistics section.

Reviewer: 1

Competing interests of Reviewer: None declared

Reviewer: 2

Competing interests of Reviewer: None

Reviewer: 3

Competing interests of Reviewer: I have no competing interests

Reviewer: 4

Competing interests of Reviewer: I understand the above, and I consent to the named publication of my review. I have no competing interests.

VERSION 2 – REVIEW

REVIEWER	Buhler, Miranda University of Otago, School of Physiotherapy
REVIEW RETURNED	22-Mar-2024

GENERAL COMMENTS	Thank you to the authors for responding to each of the reviewer comments. The manuscript is greatly improved. The one remaining concern is in relation to completeness of discussion of any potential bias. For the limitations that may introduce bias, please give your estimation (with reason) of how each limitation might influence the results (i.e., over or underestimate the finding, or if entirely unclear then again why this is so). Just stating that there may be a bias is making the reader do all the work. Otherwise a couple of very minor comments are: - Please give a ref for assumed gender differences in work-related vibration exposure grip type RE: “Also, it is possible that vibration exposure in women more often comes from precision instruments, such as dental instruments that require a pinch grip,” - Check the apostrophes for “much” RE: “Tasks requiring much pinch strength have been associated with an increased risk of hand OA .(13)..”
--

REVIEWER	Boutsikari, Eleni National and Kapodistrian University of Athens, Medicine
REVIEW RETURNED	17-Feb-2024

GENERAL COMMENTS	No further comments.
----------------------

VERSION 2 – AUTHOR RESPONSE

Reviewer: 4

Ms. Eleni Boutsikari, National and Kapodistrian University of Athens

Comments to the Author:

No further comments.

Response: Thank you!

Reviewer: 3

Dr. Miranda Buhler, University of Otago

Comments to the Author:

Thank you to the authors for responding to each of the reviewer comments. The manuscript is greatly improved. The one remaining concern is in relation to completeness of discussion of any potential bias. For the limitations that may introduce bias, please give your estimation (with reason) of how each limitation might influence the results (i.e., over or underestimate the finding, or if entirely unclear then again why this is so). Just stating that there may be a bias is making the reader do all the work.

Response: We have added comments concerning the response rate, the lack of inclusion of cases with milder symptoms and disability, information about dominant hand and certain socioeconomic status and how they may potentially impact the findings as over- or underestimations.

Otherwise a couple of very minor comments are:

- Please give a ref for assumed gender differences in work-related vibration exposure grip type RE: "Also, it is possible that vibration exposure in women more often comes from precision instruments, such as dental instruments that require a pinch grip,"

Response: We have added several references that may be related to the gender issues and exposures in occupations as well as made a slight modification of the text.

- Check the apostrophes for "much" RE: "Tasks requiring much pinch strength have been associated with an increased risk of hand OA .(13).."

Response: Text modified.

VERSION 3 – REVIEW

REVIEWER	Buhler, Miranda University of Otago, School of Physiotherapy
REVIEW RETURNED	22-May-2024
GENERAL COMMENTS	No further comments, thank you